# Transmission of images through the underwater acoustic channel to submerged networks

Alice Danckaers
ENSTA Bretagne
alice.danckaers@gmail.com

Mae L. Seto, Ph.D., P.Eng., SMIEEE
Defence R&D Canada
mae.seto@dal.ca

*Abstract* — As an acoustic communications medium, water is characterized by frequency dependent attenuation, short range, very low bandwidth, scattering, and multi-path. It is generally difficult to acoustically communicate even terse messages underwater much less images.  For the naval mine counter-measures mission, there is value in transmitting images, if possible.  The contribution of this paper is a methodology to encode, compress and transmit small (100s of kB) side-scan sonar images underwater with micromodems.  The work has been validated through several at-sea trials. The reconstruction of the received images is of a fidelity that operators can visually recognize targets they normally look for in these images.

*Keywords- underwater communications, transmission of sonar images underwater, image compression, image encoding*

## I. INTRODUCTION

Communications between robots and other systems are especially challenging underwater. Communications is needed for mobile networked underwater robots to communicate with operators, other robots, or submerged networks in a collaborative mission.  The example mission here is the naval mine counter-measures (NMCM) one with collaborating unmanned underwater vehicles (UUV) and unmanned surface vehicles (USV).  The mission is an underwater survey with a side-scan sonar (SSS) integrated on UUVs with in situ processing on the vehicle's payload processor to detect targets in the sonar data stream. The result of such processing is an image snippet from the sonar data stream containing a target (e.g. a mine-like object, MLO) and a text file with the target's geo-referenced location.  The geo-referenced location makes it possible to revisit targets to confirm their location or for other follow-on actions [1]. The image is usually viewed by the operator after the UUV is recovered.   Some NMCM operations require an operator evaluate the target prior to subsequent actions for its classification or disposal.

The NMCM mission can be more efficient with multiple UUVs due to the linear increase coverage rate with increase SSS sensors in the water. Multiple UUVs confers some redundancy when an UUV is unable to complete its mission. Exploiting the redundancy potential is only possible if the UUVs on the network have been communicating their status (position, waypoints acquired, velocity, heading, energy left) as well as geo-referenced target locations.  Such underwater communications relayed above-water also make it possible for the operator to monitor the mission.   However, underwater communications are problematic at best.

Above water, electromagnetic (radio) and light waves provide more than adequate bandwidth for transmission of sonar images that are 100s of kB in size.  However, these signals are rapidly attenuated underwater.  Acoustic signals are most practical for range and bandwidth for underwater communications.  Even with acoustic signals, underwater bandwidth and range are still limited as typical carrier frequencies are only tens of kHz (< 50 kHz) vice MHz and GHz for in-air communications above-water.

Whether the internet protocol (IP) should be TCP or UDP depends on the application.  With TCP, packet transmissions are reliable and ordered at the cost of more overhead (tracking, socket connections, etc.). A packet is re-broadcasted until there is acknowledgement it was received.  Such acknowledgement can take seconds or more (especially in poor propagation conditions) which ties up the channel.  This however is necessary for a zipped file as it cannot be reconstructed without every bit of the file reliably received.

With UDP there is no acknowledgement transmissions are received, no ordering (later transmitted packets can arrive before previous ones) therefore, it is lightweight.  Message quality is only checked at the receiving end.  With UDP it is also possible to broadcast simultaneously to more than one receiver.   The choice of UDP means message acknowledgement is a lower priority. For example, UDP can be used to transmit video as a dropped packet means a poorer quality image but an image can be reconstructed.  UDP uses limited bandwidth, which is the case of underwater acoustics, more efficiently.   The choice of UDP-like communications underwater is motivated by the high latency and low bandwidth of the environment.  The result is better efficiency.

UDP micromodems employ additional encoding on the binary packets to limit data loss.  Checksums are used in the packets.  An ACK bit which, if enabled, is like TCP in that a packet is re-transmitted until there is acknowledgement that it was received.  However, this can tie up a channel for a packet that may not be received for whatever reasons.

Traditionally, the UUV would be recovered and the sonar data and/or images downloaded for the operator to analyze. This can take a fair amount of time depending on the area to survey and the UUV cannot be re-tasked insitu to revisit a target or task another to revisit the target.   With modern NMCM missions, the timeliness is important.  What is desired is a way to transmit such images underwater from the UUV through the underwater path to an operator or another system. Therefore, the UUV does not have to be recovered for the operator to evaluate the target(s).  This has never been done before and is the objective of the project described in this paper.

The quality of underwater acoustic communications is further undermined by scattering and multi-path as well as locally and temporally variable sound speed profiles. It is generally difficult to acoustically communicate even terse messages underwater much less images. However, for the NMCM mission there is value to transmitting images off the UUV to operators. How this is achieved is addressed next.

The contribution of this project is a methodology to encode, compress and transmit small (100s of kB) SSS images underwater with micromodems. The reconstruction of the received images is of a fidelity that operators can visually recognize targets they normally look for in these images.

The system developed for this project is a transmitter (UUV) and a receiver (operator) unit. Each unit consists of an underwater micromodem serially tethered to, and deployed from, a computer. The transmitter computer is a payload processor used on-board the UUV. The computer on the receive unit is the one the operator normally uses to monitor an underway UUV mission from a RHIB, jetty, or support ship with a deck micromodem. The receiver unit also has an in-air radio connected to its computer to broadcast the receiver messages to multiple above-water locations (for prototyping purposes).

This paper reports on this project and includes: a review of potential compression algorithms; a trade-off analysis of several potential compression algorithms; realizing the selected methodology as ROS nodes; initial end-to-end algorithmic validation in an indoor water tank in a controlled environment; at-sea testing in a real underwater environment; integration of the ROS nodes with an automated target detection (ATD) system, and final in-water testing and evaluation at the Unmanned Warrior 2016 exercise.

To achieve the objective, the first task is to select a compression algorithm appropriate for SSS type images.

II.    SELECTION OF IMAGE COMPRESSION ALGORITHM

This section describes the micromodem used for the underwater communications as it affects the transmit/receive protocols as well as disposition of the messages and consequently, drives the compression algorithm requirements. Then, several algorithms are compared and evaluated and the most promising one selected for implementation.

A.    WHOI Underwater Acoustic Micromodems

One way the underwater community adapts to the difficult communications medium is through terse communications with small acoustic packets. The project reported here uses the example of the Woods Hole Oceanographic Institution

TABLE I.        WHOI MICROMODEM TRANSMISSION RATES

| data rate | frame size | frames / packet | packet size (bytes) |
|---|---|---|---|
| FSK rate 0 | 32 | 1 | 32 |
| PSK rate 1 | 64 | 3 | 192 |
| PSK rate 4 | 256 | 2 | 512 |
| PSK rate 5 | 256 | 8 | 2048 |

*This work was sponsored by Defence R&D Canada and ENSTA Bretagne*

(WHOI) underwater acoustic micromodem [2]. These micromodems are used on-board UUVs, USVs (as underwater to above-water relays), and stationary submerged seabed sensors for communications.

These modems can be polled (master-slave) for communications or through random access. Messages are available to provide status updates. The interface is NMEA (National Marine Electronics Association) compatible so the commands and status messages are ASCII. Binary data transfers are handled by hex-encoding within an NMEA sentence. All transmitted data is executed as binary fixed-length packets. Within a network of collaborative UUV or USV, any member of the network can transmit at any time. Monitoring of the network and queries for status or sensor information can come from a relay with the operator in the loop. As an example, vehicles may be launched and then queried for status at mission start then continuing to interact asynchronously. Consequently, with many UUVs in the water the likelihood of one system interfering with another increase. The data rates possible for these modems are shown in Table 1. The reliability of the transmission varies inversely with the packet size, i.e. the lowest transmission rates are most reliable. With the micromodem capabilities defined, the image compression algorithm can be considered. The compression methodology must accommodate very small packet sizes.

B.    Literature Review of Image Compression Algorithms

The merits of four compression methods were compared and evaluated for application to SSS images as there are no specific techniques that the community uses.

Common compression methods like JPEG [3] are not favored for this application as they are lossy so parts of the image become unrecoverable in the compression. JPEG2000 [4] has recoverable loss but does not compress as well. It will be considered here as a relatable reference for comparison.

Huffman coding [5] is another method applied to images. It is based on frequently-appearing values having shorter bit representations and less common values would have longer ones. To achieve this, a tree represents the priority of each value based on its frequency occurrence. A given value is coded through a traversal of the tree. Consequently, Huffman coding is more efficient than .zip or .rar compression.

Reference [6] developed a compression method specific to seafloor images by exploiting the redundancy in them. An image is deconstructed into a mosaic of constituent tiles. Each constituent tile is compared against tiles cataloged in a common database at the transmit and receive end. Once the most similar tile in the database is identified for a constituent tile, that tile's database index is retained (and transmitted) as representative of that part of the image. The tile similarity is assessed through the Euclidean distance between a constituent and database tile. This process is facilitated through a principal component analysis (PCA) [7]. At the receiver end when all the indices for all constituent tiles are received, the image can be reconstructed from the common database.

The fourth method considered for SSS images was SPIHT (Set Partitioning in Hierarchical Trees) [8]. SPIHT uses

wavelet transforms to encode images. The transmission is based on two steps: to order the wavelet coefficients byThe transmission is based on two steps: to order the wavelet coefficients by magnitude and then to transmit the most significant bits, first. The encoder (transmit end) and decoder (receive end) have a common ordering scheme so it is not transmitted. A spatial orientation tree defines the data ordering based on a recursive four sub-band splitting [8]. The tree is defined so each node has either no offspring (the leaves) or four offspring (Figure 1). Consequently, the compressed image can be truncated at various points and decoded to give a series of increasingly refined (higher resolution) versions of the initial image. Reference [9] applied SPIHT to compress synthetic aperture radar images (which are not unlike SSS ones). Their work focuses on image texture and improves compressibility by reducing speckle. Reference [10] implemented SPIHT for sonar images.

A detailed evaluation of the four identified methods, against a set of test sonar images, is described next.

## C. Evaluation of Image Compression Algorithms

The evaluation determines which of the four algorithms best meets the compression needs for sonar images. The criteria are:

- compression ratio: this is critical as image quality can be overlooked if the targets are visually recognizable to an operator and

- compressed image distortion: measured through the peak signal to noise ratio (PSNR) – unfortunately, not consistently a measure of image distortion.

The four algorithms were initially implemented in MATLAB© for the purposes of the evaluation. The results are summarized in Table II. To create the vector quantization database, a test set of 500 SSS images were deconstructed into constituent tiles (redundant tiles removed) and inserted into the database. They include sonar images from multiple types of SSS. This resulted in a 15,992-tile (88 MB) database. This is a

TABLE II. PERFORMANCE OF FOUR COMPRESSION METHODS APPLIED TO SONAR IMAGES

| | *Method* | | | |
|---|---|---|---|---|
| | *JPEG 2000* | *Huffman coding* | *vector quantization* | *SPIHT* |
| compress ratio | 9.449 | 8.916 | 251.682 | 25.924 |
| PSNR | 29.52 | 21.66 | 18.19 | 21.94 |
| pros | constant updates | | no distorts fr compression | image can be truncated to give refined versions |
| | | | quality dep on database | |
| | previous info not required to compress | previous info not required to compress | | previous info not required to compress |
| cons | distortions under high compression | distortions under high compression | requires a tile database | distortions under high compression |

respectable size and is manageable by the payload processors on-board the UUVs.

Additionally, Figure 2 shows a visual comparison of the four algorithms' performance on a common sonar image (49×113 pixels) from the Klein 5500 towed sonar. The visual appearance is important as that is how an operator would assess whether a target was mine-like.

As shown, JPEG2000 and Huffman coding reduce the file size by less than 10 times and distorts the images. The SPIHT algorithm provides either visually satisfactory results with a poor compression ratio or, distorted images with a good compression ratio. Which, depends on the maximum number

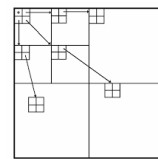

Figure 1. Example of parent-offspring dependencies in the spatial-orientation tree for the SPIHT algorithm [8].

of steps captured through the *maxloop* parameter. The best results are achieved with the vector quantization method. The compression ratio is more than satisfactory resulting in a compressed file of only 100 bytes. Furthermore, the image is visually like the original image which has value with the operators. While the PSNR could have been used as a similarity measure to the original image in this example, it was not. It was not representative enough of visual similarity.

The most promising compression scheme for underwater sonar images from the evaluation is the vector quantization algorithm [6]. This method's performance depends, not unexpectedly, on the quality of the database. The optimal database would have many tiles with great diversity between tiles that capture a variety of seabed bottom types (gravel, sand, clay, mud, etc.).

Therefore, vector quantization was selected as the image compression method and its implementation is discussed next.

## III. VECTOR QUANTIZATION IMPLEMENTATION

The results from a sensitivity analysis on parameters that affect the vector quantization compression are presented and discussed. Then, the implementation of the algorithm as ROS nodes (C++) is briefly described.

## A. Sensitivity Analysis of Vector Quantization Parameters

The vector quantization method has parameters that affect the quality of the image compression. For example, there are two thresholds in the database creation to determine whether a tile needs to be added. This, in turn, affects the execution speed and compression quality. To gain insight into these parameters' impact, a sensitivity analysis was performed. The analysis focuses on 4 parameters deemed most significant:

- number of dimensions retained in the principal components analysis

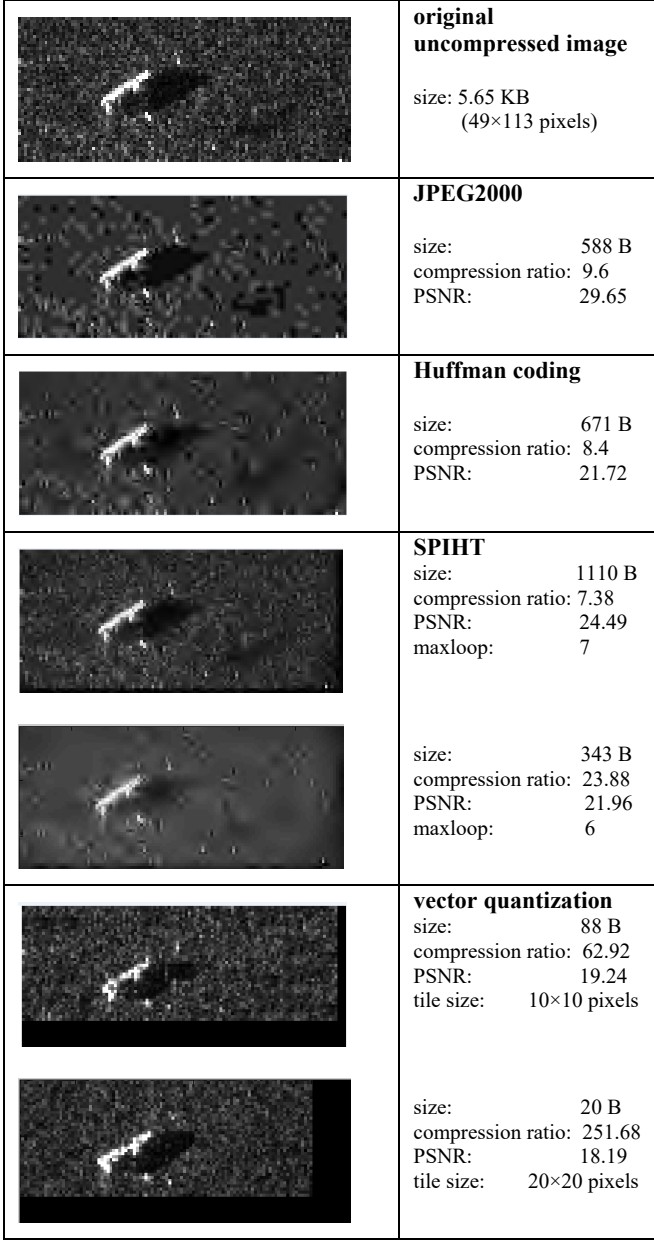

| | original uncompressed image

size: 5.65 KB
(49×113 pixels) |
| | **JPEG2000**

size: 588 B
compression ratio: 9.6
PSNR: 29.65 |
| | **Huffman coding**

size: 671 B
compression ratio: 8.4
PSNR: 21.72 |
| | **SPIHT**
size: 1110 B
compression ratio: 7.38
PSNR: 24.49
maxloop: 7 |
| | size: 343 B
compression ratio: 23.88
PSNR: 21.96
maxloop: 6 |
| | **vector quantization**
size: 88 B
compression ratio: 62.92
PSNR: 19.24
tile size: 10×10 pixels |
| | size: 20 B
compression ratio: 251.68
PSNR: 18.19
tile size: 20×20 pixels |

Figure 2. Compression algorithm comparisons for a Klein 5500 image.

- threshold for a user-defined figure-of-merit (FOM) that assesses whether an image reconstruction is adequate

- threshold for the Euclidean distance to the database that defines what tiles should be added to the database (if any), and

- number of images in the database.

The results are evaluated using a set of test images where the following were calculated: the FOM; the difference in the ATD confidence that the image contained an MLO before and after compression; the time to compress an image; the size of the database, and the time to create the nominal database (a one-time activity).

The ATD confidence is obtained through a fusion of several filters and represents the likelihood the ATD detected an MLO. Its value is a rational number between 0.0 to 2.0. The ATD confidence (not shown) was inconclusive as it is mostly constant over the test images examined. However, when the number of tiles used to create the database increases it became more diverse and the maximum error in the ATD confidence diminishes. This proves that diversity in the database ensures better reconstructed images, as expected.

Figure 3 shows that by retaining more dimensions in the PCA, the time needed to create the database as well as the time to compress an image and the size of both the rotation matrix (for the PCA) and the database, increases. But the results are visually better even with no significant difference in the FOM and the ATD confidence (not shown). Thus, the number of dimensions retained was set to 70, as it is a reasonable compromise between reconstructed image quality and computation time.

By changing the number of images in the database to improve the results, the computation time, execution time and memory required, rises quickly. The solution, then, was to build the database incrementally. First, many images would be used to generate the nominal database with a low FOM and high distance threshold. This means only tiles from images that were poorly reconstructed and are quite diverse from those in the database were added to the database. Then, the number of images and threshold distance is reduced and the FOM threshold is increased. This process was repeated multiple times to arrive at a database that is a reasonable compromise between computational efficiency and resulting image quality.

With these sensitivities in mind, the algorithm is next realized as ROS nodes.

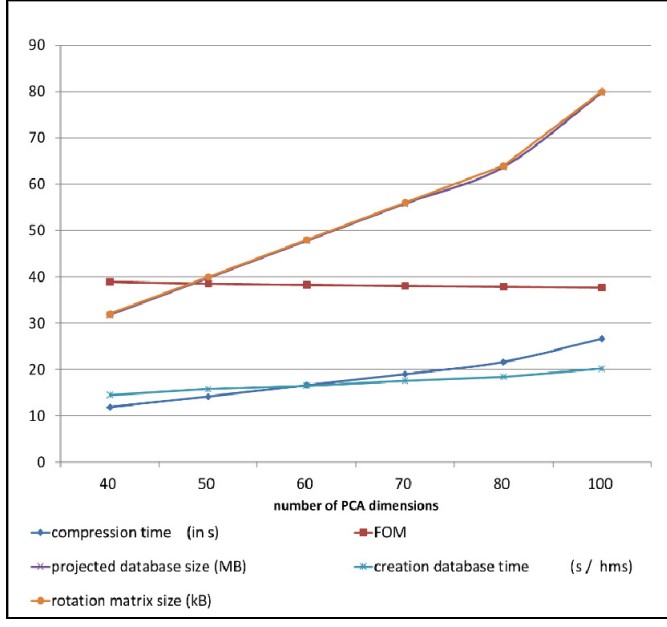

Figure 3. Sensitivity of vector quantization parameters to the principal components dimensions retained

## B. Description of Algorithm

All the code is original to the project except for the PCA which was borrowed from the Eigen Library for Matrix Algebra in C++ [7]. The developed code is organized as follows:

- **VQScheme** class contains methods and functions for image compression and decompression

- compression is achieved through *Compress.cpp* compiled as an executable (algorithm in Table III)

- similarly, decompression is achieved through *Decompress.cpp* (not shown)

- the vector quantization database is formed through *Database.cpp* and support functions (algorithm in Table IV).

**VQScheme** has one private function, *matchingTile*, which is defined for vectors of doubles or unsigned integers. This function takes as parameters a 1D-vector of 8 bit unsigned integers that represent a tile and a 2D-vector that represent a database. It is also defined for 1D-vector of doubles that represents a tile and a 2D-vector that represents a database so that it can be used for constituent and database tiles. It compares the constituent tile to those in the database and returns the index of the one that is most similar. This function is used to compress an image by the encode function. During its execution, the function affects the value of the distance between the closest tile found and the constituent tile.

## C. ROS Nodes

ROS (robotic operating system) is a widely supported open source middleware for robotic systems [11]. It provides operating system services, including hardware abstraction, low-level device control, implementation of commonly used functionality, message-passing between processes, and package management. It also provides tools and libraries to build, write, and run code across multiple platforms. It is light weight enough that it is widely applied in embedded systems. ROS works as a publish-subscribe architecture.

ROS is used by Defence R&D Canada (DRDC) as a middleware for its IVER3 UUVs' on-board autonomy. Implementation of the compression and transmit algorithm as ROS nodes facilitates rapid integration into this autonomy.

There are nodes on the transmit (UUV) side and the receive (operator) side. On the transmit (UUV) side, the compression algorithm was consists of 3 ROS nodes described below and illustrated as an ROS rqt_graph in Figure 4.

- *talker*: Tasked with publishing the path of the images to compress on the dedicated topic *imgToCompress*. Initially, it simulates the ATD algorithm detecting MLO images from the sonar data stream. Later it was directly integrated with the ATD.

- *compressor*: Subscribes to the topic *imgToCompress*. Each time it receives a new image path it compresses the image and publishes the encoded image on the topic *imgToDecompress*.

- *uuvModem*: This node subscribes to the topic *imgToDecompress*. It is connected to the micromodem through a serial port. For each encoded image it receives,

TABLE III.    IMAGE COMPRESSION ALGORITHM

| | |
|---|---|
| 1: | **Algorithm image_compression** *(path to image or folder containing images)*: |
| 2: | *load database from binary file* |
| 3: | IF *input = folder* |
| 4: | WHILE *non-compressed images in folder* |
| 5: | *read .tiff image* |
| 6: | *VQScheme::encode* |
| 7: | *save coded image in .txt. file* |
| 8: | END WHILE |
| 7: | ELSE                              *// input = image file* |
| 8: | *read .tiff image* |
| 9: | *VQScheme:: encode* |
| 10: | *save coded image in .txt file* |
| 13: | END IF |
| 14: | RETURN *(.txt file in folder with initial image name)* |

TABLE IV.    DATABASE CREATION ALGORITHM

| | |
|---|---|
| 1: | **Algorithm create_database** *(path to database images folder)* |
| 2: | IF *database core = empty* |
| 3: | *tile first image and add to database* |
| 4: | WHILE *non-treated image in folder* |
| 4: | *read .tiff file* |
| 5: | *reconstruct image with current database* |
| 6: | *compute FOM* |
| 7: | IF *FOM < 39* |
| 8: | *tile image* |
| 9: | IF *distance to closest database tile > 320* |
| 10: | *add tile to database* |
| 11: | END IF |
| 12: | END IF |
| 13: | END WHILE |
| 12: | ELSE                 *// database core != empty* |
| 13: | *load database from binary file* |
| 14: | WHILE *non-treated image in folder* |
| 15: | *read .tiff file* |
| 16: | *reconstruct image with current database* |
| 17: | *compute FOM* |
| 18: | IF *FOM < 39* |
| 19: | *tile image* |
| 20: | IF *distance to closest database tile > 320* |
| 21: | *add tile to database* |
| 22: | END IF |
| 23: | END IF |
| 24: | END WHILE |
| 25: | END IF |
| 26: | RETURN *(decoded image as .tiff in out folder with initial image name)* |

it sends it through the underwater acoustic link using the appropriate messages.

On the receive (operator) side (Figure 5), are two nodes:

- ***opModem***: This node is connected to the receiving micromodem through a serial port. It 'listens' to all messages sent by the UUV's modem to find the packet that contains the encoded image. When that message is found, it extracts the content of the message and publishes it on the topic *imgToDecompress*.

- ***decompressor***: This node subscribes to the topic *imgToDecompress* and when a new encoded image is received, it decodes it and saves the reconstructed image to the /out folder.

How these nodes interface and communicate with the underwater acoustic modems is described in detail next.

### D. Underwater Acoustic Communication

The micromodems can transmit/receive at several discrete rates that vary with packet size. For the initial in-laboratory tests, rate 4 (Table 1) was chosen so an encoded image could fit into the minimum number of packets of one frame each. This makes it easier to perform the nodes' algorithmic proof-of-principle tests at the higher risk of losing packets. Losing the occasional packet was not a primary concern for these tests. Packet loss will be addressed in the at-sea tests. The proof-of -principle tests were performed in an indoor water tank where the transmit and receive micromodems are about 3 meters from one another and thus limits packet loss. The final at-sea implementation will be performed at rate 1 which is more reliable albeit at a cost of smaller packets which means the images would be transmitted over multiple frames. This will be explained in a later section. The maximum size of a frame at rate 4, is 256 bytes so an encoded image could be transmitted over two packets. This means half of one encoded image, at a time, would be published on the topic *imgToDecompress*. As shown in Figure 6, the message published contains a header with the first character identifying whether the packet contains the top or bottom half of an image followed by the image identifier coded on four characters to recognize the origin of the half image. The rest of the message is half of the encoded image.

The ***uuvModem*** node is connected to the micromodem through a serial port, which was configured in write-only mode. When a message is published on the topic it then informs the micromodem that a packet will be sent from one modem to another. The modem then replies with a query for the packet to send and the user then provides the hex-encoded message to the modem.

On the operator side, the ***opModem*** node configures the connection with the modem through the serial port to read and

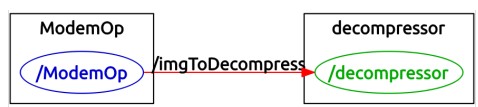

Figure 5. Interconnection of nodes on the receive (operator) unit

write mode. At initialization of the node, it will set the identifier for the operator's modem to '1' using a similar command to (1) above. Then, the node listens to the modem output through the serial port. If it read a CARXD message it means a packet was received. The content of that packet is read and published on the topic *imgToDecompress*.

The ***decompressor*** node is subscribing (listening) to the topic *imgToDecompress*. When a message is published on that topic, this node will identify whether the code is for the top or bottom half of an image and its identifier by reading the message header. If it previously received the other half of the image, it will concatenate the two and decompress the image using functions from the **VQScheme** class. But, if the other half came from a different image, then the older packet will be decompressed after zero padding its missing half. The last packet received will be stored to wait for the other half for decompression. If no other half was received, the node will wait for the next packet for decompression.

### IV. PRELIMINARY VALIDATION IN A WATER TANK

After the ROS nodes were developed, they were tested in an indoor water tank with two submerged micromodems each serially connected to a computer − the transmit (UUV) and receive (operator) units. The transmit (UUV) unit hosted the ***talker*** node. The transmit unit simulates the ATD detecting MLOs in the sonar stream and its modem simulates the on-board UUV modem. The receive (operator) unit is used by the operator to receive the compressed images.

During these tests, the communications code had to be adapted so that it could handle broken packets (a very real situation). Previously it could deal with a missing packet but if a packet was received it was considered to contain all the data needed to reconstruct half the sonar image. However, because of the highly reverberant water tank (small size and lack of acoustic cladding) some of the acoustic packets would be damaged: the end of a packet might be unintentionally padded with zeros or random numbers. The code was adapted so that if a tile index in the encoded message was greater than the maximum index possible then the data was considered corrupted in the transmission and replaced with a black tile.

After these adaptations, the tests were successful. A few packets were lost or damaged because of the tank reverberations and rate 4, expectedly, is not as reliable as rate 1. These tests validate the implementation of the compression/decompression algorithms and the transmit/receive protocols used. They also suggest that for operational conditions, the micromodems use rate 1. The at-sea tests are described next.

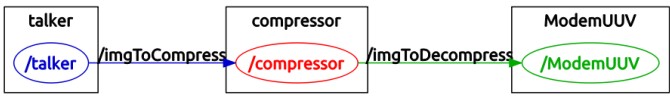

Figure 4. Interconnection of nodes on the transmit (UUV) unit

## V. Initial Test and Validation At-Sea

The main objective at-sea tests was to assess the transmission and reception success over larger micromodem separations, in a body of water with varying sound speeds, and real seabed. All these factors present underwater communication challenges to successful message transmission and reception. Otherwise, the testing and validation in Bedford Basin (to depths of 20 m) were like those in the indoor water tank. The same images would be transmitted and received at varying ranges to assess the communications impact of range.

The transmit (UUV) unit computer was on-board a RHIB with the modem deployed over the side to 6.4 m depth. On the receive (operator) unit, the micromodem was tethered to a barge at the same depth. However, the receive computer was also in the same RHIB and the communications with the operator's modem was carried out by in-air radio. Having the operator's computer on-board makes no difference functionally. However, it facilitated insitu and immediate analysis of what the operator would receive and allowed for insitu adaptation of the test matrix. The underwater transmissions were at ranges of 2 to 300 m between UUV and operator units. Sample results are presented in Figure 6.

At less than 130 m range between the transmit and receive units, given the acoustic propagation conditions that day, most transmitted packets were received. The transmission was 100% successful when the size of the images were 140 × 80 pixels. For smaller images the second packet were missing in some cases. This error was inconsistent and might be due to the second and third frame of the packets being empty or not full to start, thus resulting in unreliable transmissions. At a greater range than 130 m most packets were lost. Finally, the compression ratio achieved in these tests is different than presented in the earlier literature survey. The indices of the

tiles were re-coded on 5 bytes instead of 3 due to the size of the database which increases the size of the compressed file. The compression achieved, however, is still satisfactory. Next, is to integrate the ROS nodes into a stand-alone node with the ATD. This provides a ROS node for a function that can be directly integrated into the UUV payload autonomy.

## VI. At-Sea Developments at Unmanned Warrior

Unmanned Warrior 2016 (UW16) was a Royal Navy hosted exercise at the British Underwater Test and Evaluation Centre in Loch Alsh, Scotland. This exercise achieved significant milestones and world firsts in collaborative robotics for mine counter-measures. The results of this R&D project with underwater transmission of sonar images contributed to this.

The project's UW16 objective was to create an UUV-in-a-box (UIB) realized as an overarching stand-alone ROS Indigo node. It calls and uses all the nodes described earlier. Above and beyond what has already been described to this point, the UIB node needed to be developed. UW16 was also a venue for experienced UUV operators to view the reconstructed sonar images to see if they could visually recognize targets within them. This is a test of the fidelity of the reconstructed image. The full AIB node was finalized, tested and demonstrated at UW16.

### A. Integration with on-board Automated Target Detection

The AIB node runs as a stand-alone ROS node that would eventually be integrated into the UUV's on-board autonomy as a function. It polls a directory for a new SSS data file (1000 pings long). Once a new sonar image file is put in the polled directory, the AIB node would call the ATD to process the file to detect any targets within it. If targets are detected, their images are encoded, compressed and the messages formed. These messages must be queued to fit amongst other regular, and unplanned, underwater communications with the operator monitoring the mission. This is how it works in the UUV on-board autonomy framework.

### B. Evaluation and Demonstration in Scotland

At UW16 the AIB node read in and processed pre-logged sonar files. The files were automatically deposited in the polled directory on a schedule that was consistent with what happens on-board the UUV. These pre-logged sonar files were collected over the duration of UW16. At UW16, the receiver unit was an underwater acoustic modem deployed from the UK RN Hazzard surface vehicle. It receives the packet, decompresses then decodes it. Then, an on-board operator analyzes the image and decides if it is an MLO. This requires the image fidelity be close to the original sonar image generated by the ATD. The decompressed and decoded images were presented to UUV and sonar operators. In an average of 18 out of 20 images, operators correctly recognized them as MLO or not. The sonars that contributed images were the MarineSonics HDS and Klein 3500. They were payload sensors on-board various REMUS 100 and IVER3 UUVs. The results of this survey showed the overall algorithm (and nominal database) used for the transmission and reconstruction of images was adequate for operators.

original image     image received at 20 m     image received at 60 m

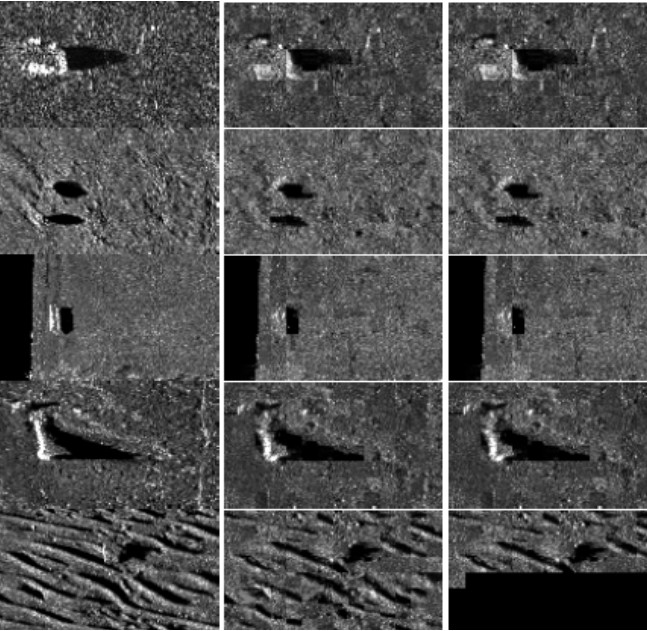

Figure 6. Results from at-sea underwater transmissions at different ranges.

Once the operator on-board HMS Hazzard examined the decompressed and decoded image, the images were further relayed, in-air, to a shore-based command and control centre. This chain of events (Figure 8) emulates the situation where the expert opinion of shore-based experts is sought in the decision with operators, for further target prosecution.

The results are encouraging. The visual images are visually quite like the original sonar image that operators would normally review and make decisions on. In separate lab tests, the compression takes around 35 seconds on the embedded processor allowing the operator to view the images while the UUV is underway and in near real-time. This makes it possible for the NMCM mission to be carried out more rapidly. The demonstration of this new capability was deemed successful.

## VII. SUMMARY DISCUSSION ON RESULTS

The algorithm appears to work quite well and is only limited by the usual underwater propagation conditions, even at rate 1. The geometry the acoustic packets formation and transmission is robust and fits within a reasonably sized number of packets.

Transmission and reception of the image snippets works well when the underwater acoustic communications channel is good for communications at 25 kHz. Not surprisingly, on days with high underwater acoustic ambient from winds and or motions of the micromodem in a sea state, packet loss occurs. The actual range for good fidelity reconstruction of images depends again on the condition of the local underwater acoustic channel. This is expected in the usual acoustic communications situation.

With the vector quantization compression, the results of the compression critically depend on the tile database diversity. Consequently, database improvement directly improves the transmitted images' quality. The database can be adapted to target specific seabed types and offer flexibility to the algorithm. Using a very large database will impact the speed that analysis occurs at. The nominal database formed in Halifax used images from a wide variety of trials and seabed types. It still performed well against the seabed types trialed in Scotland.

## VIII. CONCLUDING REMARKS

A compression method appropriate for side-scan sonar image snippets was identified, developed and implemented as ROS nodes. The compression scheme was tested on sonar images collected from a variety of sources (and hence sonars). Then, the sonar image transmissions were tested in a real-world at-sea environment. Finally, the compression and transmission scheme was integrated with ATD tools and tested, evaluated and demonstrated at the Royal Navy Unmanned Warrior 2016 exercise in Scotland.

Future engineering work on this project would integrate the nodes into the UUV autonomy framework. Further research and development work would make use of tracking methodologies to increase likelihood of reception given evolving underwater acoustic channel conditions [12].

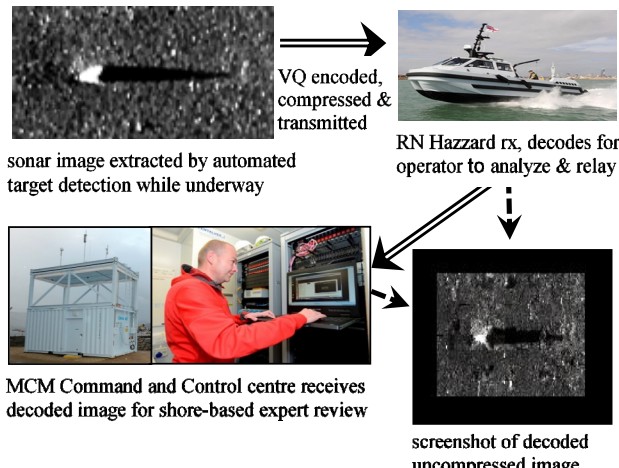

sonar image extracted by automated target detection while underway

RN Hazzard rx, decodes for operator to analyze & relay

MCM Command and Control centre receives decoded image for shore-based expert review

screenshot of decoded uncompressed image

Figure 7. Sonar image transmission at Unmanned Warrior 2016

## ACKNOWLEDGMENT

This project is grateful for the timely advice from the WHOI Acoustic Communications Group.

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
