# OpenReview forum: "Transmission of images through the underwater acoustic channel to submerged networks"
_roboticsfoundation.org/RSS/2017/RCW_Workshop/-_Proceedings_

### Review · AnonReviewer1 · 2017-06-27
**Review of "Transmission of images through the underwater acoustic channel to submerged networks"**

**Rating:** 4
**Confidence:** 2

**Review:**

This submission clearly fits the overall theme of the workshop, and presents sonar image compression and decompression algorithms that were tested on a number of platforms, demonstrating multi-agent communication in a noisy, challenging environment.  On these merits and the potential for interesting discussions, the submission should be accepted.

I have some questions that will be worth answering in the final submission and at the workshop.  While I am not an underwater communications expert, I am familiar (and I expect other attendees will be as well) with communications challenges in noisy environments with packet loss, and spacecraft communications with weak signals over long distances.
- Do the micromodems employ any additional encoding on the binary packets to limit data loss, or do they perform any additional checks on data quality (like checksums), beyond what the serial protocol itself already dues?  The serial protocol has some rudimentary checks, but in practice, wireless serial communications can be problematic without additional protocols.  If not, there is plenty of literature on how to best encode data such that the contents can be retrieved even with numerous missing/incorrect bits and a very low signal-to-noise ratio.
- Is timeliness so important that images with missing packets are preferable to using a system to ensure all packets arrive?  In Internet protocols, UDP favors speed over packet guarantees, while TCP does the opposite.  If this has not already been ruled out for underwater use, implementing TCP might be a good first attempt to handle packet loss over the micromodems.  Otherwise, please provide an explanation of why Internet protocols are not useful in underwater settings as I am sure many readers may wonder about this.
- Can the micromodems be connected in an ad-hoc network?  For significant signal attenuation, having intermediate nodes might be preferable to long packet completion times with low signal-to-noise ratios, but that is a tradeoff worth thinking about.

Formatting and text comments:
- sss should be capitalized.
- The level of specific details varies from section to section.  On one hand, this paper describes headers of serial strings (e.g. "$CCTXD...") while the algorithms are mostly textual pseudocode.  Describing the serial headers is only relevant if there is open source code available or there is a chance of someone wanting to interface with your work; it might be best to be less specific.
- Figures 4 and 5 have text overlapping boxes.
- Line spacing in Table II is inconsistent.

---

> ### Comment · ~Mae_L_Seto1 · 2017-07-15
> **Authors' response to reviewer 2**
>
> - While I am not an underwater communications expert, I am familiar (and I expect other attendees will be as well) with communications challenges in noisy environments with packet loss, and spacecraft communications with weak signals over long distances.
>
> - Do the micromodems employ any additional encoding on the binary packets to limit data loss, or do they perform any additional checks on data quality (like checksums), beyond what the serial protocol itself already dues? The serial protocol has some rudimentary checks, but in practice, wireless serial communications can be problematic without additional protocols. If not, there is plenty of literature on how to best encode data such that the contents can be retrieved even with numerous missing/incorrect bits and a very low signal-to-noise ratio.
>
> The micromodems do indeed employ additional encoding on the binary packets to limit data loss.  Checksums are used in some of the packets.  They also use an ACK bit which, if enabled, is similar to TCP in that a packet is transmitted and re-transmitted until there is acknowledgement that it has been received.  However, this can tie up a channel for a packet that cannot be received perhaps due to propagation conditions and cannot be used to broadcast anything else.
>
> This is inserted in page 1.
>
> - Is timeliness so important that images with missing packets are preferable to using a system to ensure all packets arrive? In Internet protocols, UDP favors speed over packet guarantees, while TCP does the opposite. If this has not already been ruled out for underwater use, implementing TCP might be a good first attempt to handle packet loss over the micromodems. Otherwise, please provide an explanation of why Internet protocols are not useful in underwater settings as I am sure many readers may wonder about this.
>
> Whether the internet protocol (IP) should be TCP or UDP is depends on the requirements and priorities.
>
> With TCP IP, the packet transmissions are reliable and ordered at the cost of more overhead (tracking, setup for socket connections, etc.). A packet is continually re-broadcasted until there is acknowledgement it was received.  An acknowledgement can take ~ seconds or more (especially if the propagation environment is poor).  This ties up a channel and makes it unavailable for other use.  This however is necessary in the case of a zipped file as it cannot be reconstructed at all without every bit of the file reliably received.
>
> With UDP there is no acknowledgement the transmissions are received, no ordering (later transmitted packets can arrive before previously transmitted ones) therefore, it is comparatively lightweight.  Message quality is only checked at the receiving end.  With UDP it is also possible to broadcast simultaneously to more than one receiver.  The choice of UDP implies that message acknowledgement is a lower priority. For example, UDP can be used for video imagery transmission as a dropped packet means a poorer quality image but an image can be reconstructed.  UDP uses limited bandwidth, which is the case of underwater acoustic coms, more efficiently.  The choice of UDP-like communications underwater is motivated by the high latency and low bandwidth which are immutable characteristics of the environment.  The result is better efficiency.
>
> This is inserted in page 1.
>
> - Can the micromodems be connected in an ad-hoc network? For significant signal attenuation, having intermediate nodes might be preferable to long packet completion times with low signal-to-noise ratios, but that is a tradeoff worth thinking about.
>
> This is a good suggestion and the community has done work towards this.  The micromodems can, and are, connectable to ad-hoc networks.  They can be networked this way to increase the range possible to overcome the high attenuation.
>
>
> Formatting and text comments:
> - sss should be capitalized.
>
> Done.
>
> - The level of specific details varies from section to section. On one hand, this paper describes headers of serial strings (e.g. "$CCTXD...") while the algorithms are mostly textual pseudocode. Describing the serial headers is only relevant if there is open source code available or there is a chance of someone wanting to interface with your work; it might be best to be less specific.
>
> Serial headers are removed.
>
> - Figures 4 and 5 have text overlapping boxes.
>
> Agreed. This is a property of the rqt_graph in ROS that generated it.  Unsure that the user has control over this.
> - Line spacing in Table II is inconsistent.
> Fixed.

---

### Review · AnonReviewer4 · 2017-06-29
**Paper describes a compression and communication mechanism to transmit underwater images using acoustic radios.**

**Rating:** 5
**Confidence:** 1

**Review:**

+ Good review of compression algorithms, their application to underwater images, along with pros and cons
+ Nice description of the system requirements and system built

- Some of the terms are unclear. For example, what are "sss images"? I initially thought that was a typo but it was used a couple of times in the first two pages
- While the end-end system is presented, there is little discussion of the interaction of the communication modality and the compression algorithm. For example, do the large losses as well as the latency in communication via the acoustic medium affect the image transmission and the intended applications? This would be an interesting topic for future study
- The intended end use of the images as presented in this paper seems to be that a user looks at them and takes some decisions. This is quite qualitative and could merit a user study with several operators looking through the original image as well as the image reconstructred from the encoding to see if there is a perceivable difference (in some statistical sense) for operators.

---

> ### Comment · ~Mae_L_Seto1 · 2017-07-15
> **Authors' response to reviewer 1**
>
> - Some of the terms are unclear. For example, what are "sss images"? I initially thought that was a typo but it was used a couple of times in the first two pages
>
> sss was described in the first paragraph.
>
> - While the end-end system is presented, there is little discussion of the interaction of the communication modality and the compression algorithm. For example, do the large losses as well as the latency in communication via the acoustic medium affect the image transmission and the intended applications? This would be an interesting topic for future study.
>
> Yes, the point is that the high attenuation and attenuation of acoustic signals underwater do affect the image transmission.  Unprocessed, the images are too large to transmit any distance.  This impacts the application of sending a reasonable quality image for an operator to assess.  This was an aspect that was not studied in much detail.  More insight on the interactions would be useful towards fine tuning the scheme used.
>
> - The intended end use of the images as presented in this paper seems to be that a user looks at them and takes some decisions. This is quite qualitative and could merit a user study with several operators looking through the original image as well as the image reconstructred from the encoding to see if there is a perceivable difference (in some statistical sense) for operators.
>
> The reviewer is quite correct that the assessment of the received messages by operators is valuable.  This is covered in Section VI of the original and revised paper.